# The Association between Laughter and Functional Dyspepsia in a Young Japanese Population

**DOI:** 10.3390/ijerph19095686

**Published:** 2022-05-07

**Authors:** Yasunori Yamamoto, Shinya Furukawa, Aki Kato, Katsunori Kusumoto, Teruki Miyake, Eiji Takeshita, Yoshio Ikeda, Naofumi Yamamoto, Katsutoshi Okada, Yuka Saeki, Yoichi Hiasa

**Affiliations:** 1Endoscopy Center, Ehime University Hospital, Toon 791-0295, Ehime, Japan; yasunori@m.ehime-u.ac.jp (Y.Y.); yikeda@m.ehime-u.ac.jp (Y.I.); 2Health Services Center, Ehime University, Matsuyama 790-8577, Ehime, Japan; kato.aki.ea@ehime-u.ac.jp (A.K.); kusumoto.katsunori.mb@ehime-u.ac.jp (K.K.); saeki.yuka.mu@ehime-u.ac.jp (Y.S.); 3Department of Gastroenterology and Metabology, Graduate School of Medicine, Ehime University, Shitsukawa, Toon 791-0295, Ehime, Japan; miyake.teruki.mg@ehime-u.ac.jp (T.M.); hiasa@m.ehime-u.ac.jp (Y.H.); 4Department of Inflammatory Bowel Diseases and Therapeutics, Graduate School of Medicine, Ehime University, Matsuyama 791-0295, Ehime, Japan; eiji@m.ehime-u.ac.jp; 5Faculty of Collaborative Regional Innovation, Ehime University, Matsuyama 790-8577, Ehime, Japan; yamamoto.naofumi.mk@ehime-u.ac.jp; 6Shikokuchuo Public Health Center, Shikokuchuo 799-0404, Ehime, Japan; okada_katsutoshi@pref.ehime.lg.jp; 7Community Health Systems for Nursing, Graduate School of Medicine, Ehime University, Toon 791-0295, Ehime, Japan

**Keywords:** laughter, friends, functional dyspepsia

## Abstract

The potential health benefits of laughter are recognized in relation to several chronic diseases. However, no study has yet investigated the association between laughter and functional dyspepsia (FD). The purpose of this study was to investigate this issue in a young Japanese population. Methods: This study was conducted on 8923 Japanese university students. Information on the frequency of laughter and types of laughter-inducing situations, digestive symptoms (Rome III criteria) were obtained through a self-administered, web-based questionnaire. Results: The percentage of respondents who laughed out loud almost every day was 64.3%. On the other hand, 1.8% of the subjects reported that they rarely laughed. No association was found between the total frequency of laughter and FD. Laughing while talking with family and friends almost every day was significantly inversely associated with FD (adjusted odds ratio (OR): 0.47 (95% confidence interval (CI): 0.28–0.81); *p* for trend was 0.003). On the other hand, laughing while watching TV or videos and laughing while looking at comics or magazines independently showed a positive correlation with FD (TV or videos: adjusted OR, 1–5 times a week: 1.74 (95% CI: 1.16–2.60); comics or magazines: adjusted OR, 1–5 times a week: 1.78 (95% CI: 1.08–2.81)). Conclusion: In this young Japanese population, no association between laughter frequency and FD was observed although laughing while talking with friends and family was independently and inversely associated with FD.

## 1. Introduction

Functional dyspepsia (FD) is widely known as a prevalent disease that reduces quality of life. A variety of factors are assumed to be associated with the development of FD: female gender, smoking, low body mass index, low physical activity, depressive symptoms, sleep disturbance, and *Helicobacter pylori* infection have been reported as risk factors for FD [1,2,3].

In other multifactorial conditions, laughter is reported to have potential health benefits. Laughter has been inversely associated with depressive symptoms [4], insomnia [5], hypertension [6], diabetes mellitus [7], poor oral health problems [8], stroke [9], heart diseases [9], and all-cause mortality [10]. Additionally, laughter might improve natural killer cell activity [11], endothelial function [12], and stress hormone levels [13] in humans. Laughter is a social activity that strengthens relationships with others [14]; meta-analyses have found an association between weak social connections and mortality and the development of cardiovascular disease [15,16].

There is limited evidence, however, regarding possible associations between laughter and digestive diseases, and there is no evidence of a link between laughter and FD. Given the known close association between depression and FD [17], however, an association between laughter and FD is not unlikely. Laughter has a protective effect against depression [4], and anti-depressive drugs have been shown to be effective against FD [18]. Therefore, we hypothesized that laughter might be inversely associated with FD.

The purpose of this study is to examine the association between the frequency of laughter and FD as defined by the Rome III criteria in young Japanese adults. The secondary purpose is to investigate possible associations between types of laughter-inducing situations, that is, social laughter versus solitary laughter, and FD.

## 2. Materials and Methods

### 2.1. Study Design

Cross-sectional study.

### 2.2. Study Population

Data from the 2015 to 2017 results of the university health examinations conducted every April were analyzed. A specific web-based questionnaire related to FD (Rome III criteria classification) was sent to all individuals who underwent the medical examination. Information on organic gastrointestinal diseases, used drugs, and physical signs to complement the Rome III criteria was also obtained using a self-administrated questionnaire. The exclusion criteria for this study were treatment for gastrointestinal diseases within the last 6 months and history of gastrointestinal disease (e.g., gastroesophageal reflux disease, gastritis, *H. pylori* sensitization, ulcers, and cancer) and liver, biliary, and pancreatic diseases; and the presence of physical symptoms related to gastrointestinal disease, such as weight loss and recurrent vomiting. The final analysis population for this study consisted of 8913 individuals, 1191 of which were excluded from the total 10,104 respondents according to the above criteria. Informed consent was obtained in the form of opt-out on the website. Those who refused were excluded. The research protocol was reviewed and approved by the Ethics Committee of Ehime University (Institutional Approval Number 1610012). In addition, the research protocol was prepared in compliance with the ethical guidelines of the Declaration of Helsinki.

### 2.3. Measurements and Definition of Functional Dyspepsia (FD)

A web-based self-administered questionnaire was used to collect information on FD-related symptoms; lifestyle habits, such as exercise, smoking, and drinking habits; and medical history. The definition of FD (Rome III criteria), current smoking, current drinking, exercise habits, and body mass index (BMI) have been addressed in detail in our previous studies [19,20].

### 2.4. Definition of Laughter

The following questions were used to obtain information about the frequency of laughter in daily life and situations that induce laughter: (1) frequency of loud laughter: “How frequently do you laugh loudly?” (almost every day, 1–5 times a week, 1–3 times a month, and almost never) and (2) types of laughter-inducing situations (multiple selections were permitted): “When do you laugh out loud?” (talking with family or friends, watching TV or videos, looking at comics or magazines, listening to the radio, and watching comic stories and/or plays).

### 2.5. Statistical Analysis

Laughter frequency was divided into four categories based on the questionnaire responses: (1) almost every day, (2) 1–5 times a week, (3) 1–3 times a month, and (4) almost never (reference). The frequency of laughter occurring in response to each type of situation (talking with family or friends, watching TV or videos, looking at comics or magazines, listening to the radio, and watching comic stories and/or plays) was also divided into four categories: (1) almost every day, (2) 1–5 times a week, (3) 1–3 times a month, and (4) almost never (reference). Crude odds ratios (ORs) and their 95% confidence intervals (CIs) were calculated using logistic regression analysis of the relationship between the frequency of laughter and FD in each type of situation. Multiple logistic regression analysis was used to adjust for age, gender, BMI, alcohol consumption, regular exercise, smoking, heart murmur, and anemia as potential confounders. Statistical analysis was performed using the statistical SAS software package version 9.4 (SAS Institute Inc., Cary, NC, USA).

## 3. Results

### 3.1. Subject Characteristics

Table 1 shows the characteristics of the study participants and the results regarding laughter. A total of 39.3% of participants reported regular exercise habits. The proportion that reported laughing loudly almost every day was 64.3%. Only 1.8% of participants reported that they rarely laughed. The most common laughter-inducing situations were talking with family or friends and watching TV or videos.

### 3.2. Association between Frequency of Laughter and FD

The association between laughter frequency and FD is shown in Table 2. The prevalence of FD was 1.7%, 2.2%, 2.6%, and 2.5% in participants who reported laughing almost every day, 1–5 times a week, 1–3 times a month, and almost never, respectively. No association between laughter frequency and FD was found.

### 3.3. Associations between Laughter-Inducing Situations and FD

Table 3 shows the results of the analysis of the relationship between FD and situations where laughter occurs. The prevalence of FD in subjects who laugh while talking with family and friends almost every day, 1–5 times a week, 1–3 times a month, and almost never was 1.7%, 2.0%, 3.0%, and 3.0%, respectively. After adjustment, laughing while talking with family and friends almost every day was independently inversely associated with FD (adjusted OR: 0.47 (95% CI: 0.28–0.81), *p* for trend = 0.003). On the other hand, laughing while watching TV or videos and laughing while looking at comics or magazines were independently positively associated with FD (watching TV or videos: adjusted OR, 1–5 times a week: 1.74 (95% CI: 1.16–2.60) and looking at comics or magazines: adjusted OR, 1–5 times a week: 1.78 (95% CI: 1.08–2.81)). On the other hand, other laughter-inducing situations were not associated with the prevalence of FD.

## 4. Discussion

In the present study, the overall frequency of laughter was not associated with FD, but the frequency of laughing while talking with friends and family was independently and inversely associated with it. On the other hand, an inverse association between laughing while watching TV or videos and laughing while looking at comics or magazines and FD was found. This is the first study to investigate the associations between particular laughter-inducing situations and FD.

Previous studies have shown that laughter has preventive effects mainly related to vascular diseases and mental disorders. Subjects with coronary heart disease and/or stroke had lower laughter frequencies compared to healthy controls in a Japanese study of 20,934 subjects [9] and in a United States (U.S.) study of 300 consecutive subjects [21]. Laughter interventions have been shown to improve postprandial hyperglycemia [22], endothelial function [12], and autonomic nervous system function [23]. The prevalence of FD, meanwhile, is related to sleep disorders [24,25,26] and psychological conditions, such as anxiety [27] and depression [4]. A meta-analysis has shown that laughter and humor interventions have preventive and ameliorative effects on poor sleep quality, depression, and anxiety in adults [28]. Similarly, a systematic review of hospital clowning in pediatric symptom management found significant reductions in stress, pain, and distress with hospital clowning [29]. In a U.S. intervention study of 18 children, humor intervention increased pain tolerance [30].

Only two studies have shown associations between laughter and digestive diseases. In patients with irritable bowel syndrome, laughter yoga was more useful for reducing gastrointestinal symptoms than anxiolytics [31]. Participants who participated in a group consultation intervention, such as a laughter yoga class, showed improvements in gastrointestinal and psychological symptoms, less frequent group participation, and fewer unnecessary invasive tests compared with participants who were given paper information including the same topics discussed in the group consultation [32]. These studies are consistent with our findings.

The underlying mechanism linking laughter and FD remains unclear. Laughter with others might be representative of health, socioeconomic status, and social bonds. Laughter with others might also be associated with positive social support. Social support might reduce cortisol and increase oxytocin levels [33]. Elevated cortisol increases visceral hypersensitivity. Elevated oxytocin may promote gastric emptying [34]. In human societies, laughter has been retained as an optimized tool for unconscious cognitive-emotional problem solving and essential social bonds preservation [35]. In a rat model, maternal separation causes gastric hypersensitivity and delayed gastric emptying [36]. In addition, acute stress in adulthood leads to colonic hyper-motility, barrier dysfunction, and visceral hypersensitivity via various neurotransmitters [37]. Laughter may have direct neuroendocrine benefits, such as reducing serum stress hormones such as cortisol and epinephrine [38]. Chronic stress induces hypersensitivity of murine gastric vagal afferents, which may contribute to the gastric hypersensitivity seen in FD [39]. Such stress amplifies the anorexic effects of cholecystokinin through activation of the nuclei that comprise the brain’s neuronal network for satiation and might play a role in the pathogenesis of the postprandial distress syndromes of FD [40].

Only laughter with friends and family was independently inversely associated with FD, and it remains unclear why different types of laughter-inducing situations have different associations with FD. Similarly, another study has shown that exercise with others but not solitary exercise is significantly inversely associated with FD [2]. Watching laughter-inducing comedy clips with close friends triggers endogenous opioid release, which significantly elevates the pain threshold in both male and female volunteers [41]. Engaging in activities with others or simply spending time with others might reduce the symptoms of FD. In this study, laughing while watching TV or videos and looking at comics or magazines was positively associated with FD. In previous studies of Japanese students, excessive Internet use was associated with depressive symptoms [42,43]. Moreover, the prevalence of sleep disturbance in students with psychiatric symptoms was higher than those without psychiatric symptoms [43]. Sleep disturbance was a well-known risk factor for FD. Excessive time spent watching TV or videos and looking at comics or magazines can shorten sleep duration and cause sleep disturbances. Hearing the punchline activates the sympathetic nervous system. Laughter as a result of hearing the punchline might cause FD via excessive sympathetic nervous system activation [44]. Thus, excessive watching of television or videos or reading of comic books or magazines can lead to sleep disturbances. The mechanisms underlying these associations, especially the relationship between laughter with others and FD, require further research.

Several limitations exist in this study. First, the data regarding depressive symptoms were based on validated questionnaires. Thus, frequency of laughter might be associated with severity of depressive symptoms. However, in the present study, no association between frequency of laughter and FD was found. Second, due to the cross-sectional nature of the study, the causal relationship between laughter practices and FD is unknown. Laughter with others might be representative of health status. Third, medical records that include information on endoscopy, *H. pylori* infection, and other previous medical history were lacking in this study (although we excluded subjects with serious physical symptoms, self-reported digestive disease, and treatment for gastrointestinal disease). Fourth, variable laughing situations may affect the intake of food. Food might cause FD; however, the data regarding food are lacking in this cohort. Fifth, although we investigated the laughter-inducing situations, including laughing with others, in this cohort, we did not assess social versus asocial laughing. Sixth, it is difficult to make an objective assessment of the frequency of laughter and the occasions of laughter. Therefore, some misclassification bias may have occurred. Seventh, we had no data on stress hormone levels in this cohort. Finally, the subjects in the current study consisted solely of a young Japanese population.

## 5. Conclusions

Although we could not find an association between frequency of laughter and FD in this young Japanese population, we did find evidence suggesting that laughing while talking with friends and family is independently inversely associated with FD.

## Figures and Tables

**Table 1 ijerph-19-05686-t001:** Clinical characteristics of 8923 study participants.

	Total (*N* = 8923)
Age, years, mean ± SD	20.1 ± 2.8
Sex, male/female	5478/3445
BMI	21.35 ± 3.05
Smoking (%)	527 (5.9)
Drinking (%)	973 (10.9)
Exercise habit (%)	3508 (39.3)
Medical history	
Heart murmur (%)	47 (0.5)
ECG abnormality (%)	63 (0.7)
Anemia (%)	239 (2.7)
Traffic accident (%)	115 (1.3)
Sport injury (%)	273 (3.1)
Frequency of laughing loudly	
Almost every day, *n* (%)	5741 (64.3)
1–5 times a week, *n* (%)	2746 (30.8)
1–3 times a month, *n* (%)	273 (3.1)
Almost never, *n* (%)	163 (1.8)
Laughter-inducing situations	
Talking with family or friends, *n* (%)	8315 (63.2)
Watching TV or videos, *n* (%)	5599 (62.8)
Looking at comics or magazines, *n* (%)	2196 (24.6)
Listening to the radio, *n* (%)	420 (4.7)
Watching comic stories and/or plays, *n* (%)	224 (2.5)
FD, *n* (%)	168 (1.9)

FD, functional dyspepsia; BMI, body mass index; ECG, electrocardiogram; SD, standard deviation.

**Table 2 ijerph-19-05686-t002:** Crude and adjusted odds ratios and 95% confidence intervals for functional dyspepsia in relation to frequency of laughing loudly.

Variable	Prevalence (%)	Crude OR (95% CI)	Adjusted OR (95% CI)
Functional dyspepsia			
Frequency of laughing loudly			
Almost every day	96/5741 (1.7)	0.68 (0.28–2.23)	0.60 (0.24–2.00)
1–5 times a week	61/2746 (2.2)	0.90 (0.37–3.00)	0.85 (0.34–2.84)
1–3 times a month	7/273 (2.6)	1.05 (0.31–4.05)	0.98 (0.29–3.82)
Almost never	4/163 (2.5)	1.00	1.00
*p* for trend			0.031

Odds ratios were adjusted for age, sex, body mass index, drinking, smoking, regular exercise, heart murmur and anemia. OR, odds ratio; CI, confidence interval.

**Table 3 ijerph-19-05686-t003:** Crude and adjusted odds ratios and 95% confidence intervals for functional dyspepsia in relation to type of laughter-inducing situation.

Variable	Prevalence (%)	Crude OR (95% CI)	Adjusted OR (95% CI)
Talking with family or friends			
Almost every day	92/5536 (1.7)	0.55 (0.34–0.95)	0.47 (0.28–0.81)
1–5 times a week	50/2510 (2.0))	0.67 (0.39–1.18)	0.59 (0.35–1.06)
1–3 times a month	8/269 (3.0)	1.01 (0.41–4.69)	0.90 (0.36–2.04)
Almost never	18/608 (3.0)	1.00	1.00
*p* for trend			0.003
Watching TV or videos			
Almost every day	64/3618 (1.8)	1.17 (0.81–1.70)	1.05 (0.72–1.51)
1–5 times a week	48/1774 (2.7)	1.81 (1.21–2.69)	1.74 (1.16–2.60)
1–3 times a month	5/165 (3.0)	2.03 (0.70–4.69)	1.96 (0.67–4.57)
Almost never	51/3366 (1.5)	1.00	1.00
*p* for trend			0.60
Looking at comics or magazines			
Almost every day	32/1385 (2.3)	1.39 (0.92–2.04)	1.35 (0.89–1.96)
1–5 times a week	21/718 (2.9)	1.77 (1.08–2.78)	1.78 (1.08–2.81)
1–3 times a month	2/59 (3.4)	2.07 (0.34–6.73)	2.24 (0.36–7.47)
Almost never	113/6761 (1.7)	1.00	1.00
*p* for trend			0.04
Listening to the radio			
Almost every day	4/275 (1.5)	0.77 (0.23–1.82)	0.80 (0.25–1.92)
1–5 times a week	3/130 (2.3)	1.22 (0.30–3.28)	1.38 (0.34–3.73)
1–3 times a month	0/15 (0.0)	NA	NA
Almost never	161/8503 (1.9)	1.00	1.00
*p* for trend			0.82
Watching comic stories and/or plays			
Almost every day	1/171 (0.6)	0.30 (0.02–1.37)	0.32 (0.02–1.43)
1–5 times a week	1/43 (2.3)	1.23 (0.07–5.70)	1.40 (0.08–6.68)
1–3 times a month	1/10 (10.0)	5.75 (0.31–30.08)	5.33 (0.29–29.54)
Almost never	165/8699 (1.9)	1.00	1.00
*p* for trend			0.32

Odds ratios were adjusted for age, sex, body mass index, drinking, smoking, regular exercise, heart murmur, and anemia. OR, odds ratio; CI, confidence interval; NA, not available.

## Data Availability

The datasets used and/or analyzed during the current study are available from the corresponding author on reasonable request.

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
