# Peer review of "The Association between Laughter and Functional Dyspepsia in a Young Japanese Population"

_ijerph, 2022, doi:10.3390/ijerph19095686_

Round 1

Reviewer 1 Report

The manuscript "The association between laughter and functional dyspepsia in a young Japanese population" by Y. Yamamoto et al. develops an interesting statistical study on laughter and FP in a significant population of yourg students (a total of 10140 respondents). One of the limitations observed by the authors is the very unreliable estimation of their main variable (frequency of laughter). For a comparison, I would suggest consulting Hasan & Hasan (2009) on a study of medical records involving laughter frequency ("Laugh yourself into a healthier person... " Int.J.Med.Sci. 6). This lack of distinction in the  main variable may be involved in the relatively weak results obtained. Nevertheless, the study is valuable and interesting. Concerning some of the curious outcomes found in the different contexts of "social" laugh (versus say, "asocial" laugh) the authors are recommended Marijuán et al., (2019), on "quantitative traits of the sociotype", BioSystems 180, 79-87. Laughter becomes tightly linked to social bonds (Navarro et al., "laughing bonds" 2016 in Kibernetes 45 (8) 1292-1307).  Note: the authors are not requested to cite these references, just to check them whether they might be useful to reinforce their discussion and enrich the article. I think they may help to explain those curious outcomes.

Some minor typos found:

Table 3, the titles of the different variable situations are placed irregularly.

Line 165, the IBS acronym lacks its referent in the text.

Line 196, might be associated

Line 198, were found

Reviewer 2 Report

This is a fine. well-written study. My only suggestions are to mention some theoretical qualifications, to wit: It is unlikely that the physiological changes entailed in laughter would by themselves be salutary. Laughter is energetically costly, so it is unlikely to abet healing. Hearing the punchline activates the sympathetic division (Langevin & Day, 1972), rather than being relaxing--and the more amusement, the greater the activation. If it were healthful to relax, why wouldn't we simply...relax? It is likely that humor and its specific expression evolved for some reason other than health (see Weisfeld, 1993). Also, people who laugh a lot may tend to be healthy simply because they worry less about their health; you need to be free of anxiety in order to be receptive to humor. In this context, it might be noted that "dyspeptic" refers to being grouchy or irritable. This too might be adaptive, leading us to avoid intake of foods that aggravate our gut. 

The finding that the benefits of laughter occurred only in the social situation is noteworthy. As the authors suggest, positive social contact of all sorts tends to lower cortisol. This is at least partly due to a rise in oxytocin, the social bonding hormone. It makes adaptive sense to relax when sources of social support and protection are present.   
